# Neurocognitive and Neuropsychiatric Sequelae in Long COVID-19 Infection

**DOI:** 10.3390/brainsci14060604

**Published:** 2024-06-14

**Authors:** Marta Almeria, Juan Carlos Cejudo, Joan Deus, Jerzy Krupinski

**Affiliations:** 1Department of Neurology, Hospital Universitari MútuaTerrassa, 08221 Terrassa, Spain; jkrupinski@mutuaterrassa.es; 2Cognitive Impairment and Dementia Unit, Hospital Sagrat Cor, Hermanas Hospitalarias, 08760 Martorell, Spain; 3Clinical and Health Department, Psychology Faculty, Autonomous University of Barcelona, 08193 Bellaterra, Spain; 4MRI Research Unit, Department of Radiology, Hospital del Mar, 08003 Barcelona, Spain; 5Faculty of Science and Engineering, Department of Life Sciences John Dalton Building, Manchester Metropolitan University, Manchester M15 6BH, UK

**Keywords:** long COVID-19, neurocognitive sequelae, persistent symptoms

## Abstract

**Objective**: To characterize the cognitive profile of long COVID-19 subjects and its possible association with clinical symptoms, emotional disturbance, biomarkers, and disease severity. **Methods**: We performed a single-center cross-sectional cohort study. Subjects between 20 and 60 years old with confirmed COVID-19 infection were included. The assessment was performed 6 months following hospital or ambulatory discharge. Excluded were those with prior neurocognitive impairment and severe neurological/neuropsychiatric disorders. Demographic and laboratory data were extracted from medical records. **Results**: Altogether, 108 participants were included, 64 were male (59.25%), and the mean age was 49.10 years. The patients were classified into four groups: non-hospitalized (NH, n = 10), hospitalized without Intensive Care Unit (ICU) or oxygen therapy (HOSPI, n = 21), hospitalized without ICU but with oxygen therapy (OXY, n = 56), and ICU (ICU, n = 21) patients. In total, 38 (35.18%) reported Subjective Cognitive Complaints (SCC). No differences were found considering illness severity between groups. Females had more persistent clinical symptoms and SCC than males. Persistent dyspnea and headache were associated with higher scores in anxiety and depression. Persistent fatigue, anxiety, and depression were associated with worse overall cognition. **Conclusions**: No cognitive impairment was found regarding the severity of post-COVID-19 infection. SCC was not associated with a worse cognitive performance, but with higher anxiety and depression. Persistent clinical symptoms were frequent independent of illness severity. Fatigue, anxiety, and depression were linked to poorer cognitive function. Tests for attention, processing speed, and executive function were the most sensitive in detecting cognitive changes in these patients.

## 1. Introduction

SARS-CoV-2, causing COVID-19, primarily affects the respiratory tract, but mounting evidence suggests its ability to cause central nervous system (CNS) injury [1,2,3,4]. Neurological symptoms (NSs) like headache, anosmia, dysgeusia [5,6,7,8,9], and cognitive complaints [10] are common in SARS-CoV-2 infection. While anosmia and dysgeusia often resolve in the acute phase [11], cognitive impairment may persist [12]. Some individuals continue experiencing post-COVID-19 symptoms, but the majority fully recover in a few weeks [13]. The World Health Organization (WHO) defines post-COVID-19 or long COVID-19 as new or lasting symptoms emerging 3 months after SASR-CoV-2 infection, lasting at least 2 months, irrespective of the symptoms’ severity or age [14]. A systematic review by Michelen et al. [15] of 39 studies highlighted various risk factors for long COVID-19, attributed to differences in study methods, sample sizes, and follow-up times, challenging syndrome comprehension.

Up to 50% of COVID-19 patients have lasting symptoms [16,17,18]. In total, 45% experience persisting fatigue after 4 months [17], coexisting with dyspnea, sleep disturbances, neuropsychological impairment, and neuropsychiatric consequences [19,20,21,22,23]. Post-viral fatigue syndrome [24,25] concerns have arisen, particularly in women and irrespective of the severity of COVID-19. The cause remains unclear [26,27,28], but could involve psychological and inflammatory factors [26,29]. A meta-analysis of 45 studies [30] identified common NSs persisting after four weeks post-infection: fatigue, neurocognitive deficits, altered taste and smell, paresthesia, headaches, and nausea. A longitudinal study six months post-COVID-19 revealed fatigue or weakness (63%), sleep disturbance (26%), anosmia (11%), dysgeusia (7%), myalgias (2%), headache (2%), anxiety (23%), and depression (27%) [21]. Study monitoring of post-COVID-19 patients reported frequent symptoms six months after hospitalization: fatigue, memory/concentration disturbances, sleep problems, and myalgias [31]. In Lopez-Leon et al.’s [32] systematic review and meta-analysis of 47,910 patients, 80% developed one or more persistent symptoms in long COVID-19, with fatigue, headache, and attention deficits being the most common.

In long COVID-19 studies, the neurocognitive deficit termed ‘brain fog’ was the most common, affecting at least one-third of patients [33,34] from moderate (58.7%) to severe impairment (18.4%) [35], surpassing the prevalence rates seen in other viral infections [29]. The most prevalent neuropsychological deficits included attention, memory, dysnomia, executive function, and speed processing [4,34,36,37,38,39,40,41]. While impacting about 86% of patients’ ability to work [34], these deficits are independent of initial disease severity [36,37,42] and are more related to mental health status [36]. Multiple neurocognitive domain impairments were more prevalent (60.3%) than a single domain (39.7%), with attention being the most commonly affected, followed by executive function [43]. Women were twice as affected as men, indicating a potential gender association [35,37,44]. Older age was linked to a higher risk of neuropsychological deficits, especially in executive function, along with a lower education level [41,44,45,46] or lower premorbid intelligence [44]. Early fatigue and NSs in the first 3 weeks post-infection strongly predicted later neurocognitive symptoms [38]. Moreover, the presence or absence of neuropsychological complaints did not predict neurocognitive performance [44]. It was suggested that processing speed, phonetic fluency, and alternate attention could better distinguish patients with long COVID-19 from controls [44,47].

In summary, Perrottelli et al.’s systematic review [48] indicated widespread neurocognitive impairment across a large majority of studies [5,36,42,43,49,50,51,52,53,54,55,56,57,58,59,60,61,62,63,64,65,66,67,68,69,70,71,72,73,74], although several studies found no significant impairment in COVID-19 patients [37,45,75,76,77,78,79]. While most assessments used screening tests, providing valuable insights into COVID-19-related neurocognitive impairment, only a few studies employed a comprehensive neuropsychological battery. Based on literature reviews, the implementation of screening assessments and neuropsychological follow-ups consistent with disease severity is recommended [80], as well as the standardization of neuropsychological batteries and criteria for neurocognitive impairment to harmonize comparisons [81].

Nearly three years into the COVID-19 pandemic, increasing evidence shows its effects on neurocognitive and neuropsychiatric functions. This study aimed to characterize clinical and neuropsychological manifestations during the post-acute phase after COVID-19 infection.

## 2. Methods

### 2.1. Study Design and Participants

This was a consecutive case series cross-sectional study at Hospital Universitari MútuaTerrasa (HUMT) that assessed adult patients between April 2020 and July 2021. All had confirmed SARS-CoV-2 infection via positive polymerase chain reaction (PCR) from nasopharyngeal swab or serology, and were aged from 20 to 60 years. The exclusion criteria were being aged over 60 years old to avoid age-related cognitive decline, individuals with prior cognitive impairment or CNS manifestations, and severe psychiatric disorders that could potentially affect cognition. The assessment was performed at 6 months (±15 days) from hospital or ambulatory discharge after COVID-19 infection.

### 2.2. Data Collection and Definitions

Data sourced from the HUMT database underwent a retrospective electronic health record analysis. The collected information encompassed demographics data, underlying comorbidities, blood test results (including ferritin and D-Dimer), symptoms, and signs at presentation and after 6 months. Cognitive complaints were examined through an open question to the participant asking if they had noticed any cognitive change after COVID-19 infection. To assess cognitive impairment, a set of subtests were selected to create a neuropsychological battery specific for this population. Neuropsychological evaluations were performed by the same expert in neuropsychology during a one-hour session. All tests were validated in our population and are used internationally. The battery included the Test de Aprendizaje Verbal España-Complutense (TAVEC) [82], which consists of a verbal memory test in which the subject must learn a list of fifteen words over five attempts. There is an interference list, free and cued immediate recall, and a recognition subtest. We used the Visual Reproduction of the Wechsler Memory Scale–IV (WMS-IV) [83], a visual memory test in which the subject must memorize four geometric figures for ten seconds to later draw the immediate and delayed recall. In the Digits forward and Backward, the subject is asked to repeat a series of numbers in the same and in reverse order. The Letter and Numbers test evaluates complex working memory, in which the subject must order the numbers and letters provided from small to bigger and alphabetically. In the Trail Making Test (TMT) part A, the subject must join a series of numbers in increasing order, with the objective of evaluating sustained attention and processing speed, and in part B, the subject must intersperse numbers and letters in increasing and alphabetical order to evaluate alternating attention and cognitive flexibility. The Symbol Digit Modalities Test (SDMT) evaluates attention and processing speed, where the subject has to fill in some symbol boxes with the number corresponding to each one. The Stroop consists of three subtests. In the first part, the subject must read written colors in order and as quickly as possible, in the second one, they must read the color of the ink printed, and in the third one, they must avoid reading the written word and say the color of the ink, with the aim to evaluate inhibition ability. In the Phonemic and Semantic fluency test, the subject must say the maximum number of words starting with P and animals (respectively) in one minute, and the Boston Naming Test (BNT) is for the assessment of denomination, in which the subject must name sixty different images from the NEURONORMA project (NN) [84,85,86,87,88,89]. All direct scores from each test and subtest were used for the study. The scores used for the analysis and comparison between groups were the standardized notes, according to the normative data in our environment, thus correcting the effects of the subjects’ age and education. Specifically, we used the T note (PT) (the mean was 50 points with SD of 10 points). The Hospital Anxiety and Depression Scale (HAD) [90] was administered to assess symptoms of anxiety and depression.

### 2.3. Statistical Analysis

The sample was categorized by illness severity into four groups: non-hospitalized (NH, n = 10), hospitalized without ICU or oxygen therapy (HOSP, n = 21), hospitalized with oxygen therapy but no ICU (OXY, n = 56), and ICU admission (ICU, n = 21). Descriptive data on Subjective Cognitive Complaints (SCC) attendance, and initial and persistent symptoms at 6 months were collected for each group, and normality was assessed for study variables.

Inferential tests were performed to compare the cognitive performances between groups. Mean comparison tests for independent data (between the study groups) were used. For data that followed a normal distribution and met the condition of homogeneity of variance, Student’s *t* was used when the comparisons were between two categories and ANOVA when there were more than two. To compare homogeneity of variance, Levene’s F was used for Student’s *t* and ANOVA. Post hoc ANOVA contrasts were performed using the Scheffé test. For a comparison of the means of the variables that did not follow a normal distribution or with a few subjects lower than 30, the Kruskal–Wallis’s rank test and the Mann–Whitney U test were used. The effect size was assessed with Cohen’s d, with values of 0.20 for a small effect, 0.50 medium effect, and 0.80 large effect. Finally, for a comparison between proportions, the Chi square (Chi^2^) test was used. Statistical analyses was performed using R, CRAN Oficina de software libre (CIXUG) (Spanish National Research Network; http://cran.es.r-project.org/, accessed on 12 October 2020).

## 3. Results

### 3.1. Demographic and Clinical Characteristics

The study comprised 108 SARS-CoV-2 patients, 64 (59.25%) males and 44 (40.75%) females, with a mean age of 49.10 years (SD: 7.67). In total, 38 (35.18%) of the subjects referred to SCC. No differences were found regarding illness severity between SCC and non-SCC group (Chi^2^: 2.192, *p* = 0.534) (Table 1).

The persistence of symptoms at 6 months for all sample was as followed: 2 subjects (1.85%) had cough, 3 (2.77%) dysgeusia, 5 (4.6%) anosmia, 8 (7.40%) myalgia, 17 (15.74%) headache, 17 (15.74%) dyspnea, and 42 (38.88%) fatigue. Anxiety and depression as persistent neuropsychiatric symptoms (scores on the HAD scale above the cut-off point) were present in 49 patients (45.3%) and 38 (35.18%), respectively.

In total, 34.4% of men had some persistent symptom compared to 59.1% of women at 6 months (Chi^2^: 6.45, *p* = 0.011). Gender differences were also observed in cognitive complaints, being more frequently reported by women in 45.5% compared to 28.1% of men (Chi^2^: 3.43, *p* = 0.05).

The statistically significant differences were observed in relation to cognitive complaints and the persistence of symptoms, with cognitive complaints being more frequent in the group with persistent symptoms (Chi^2^: 24.66, *p* = 0.001); 73.2% of the patients did not have any, and 76.3% of the patients with persistent symptoms also had SCC.

A total of 76.31% of individuals with SCC had persistent anxiety and 63.15% persistent depression. In total, 64.28% of patients with fatigue had persistent anxiety and 54.76% depression, while 82.35% of patients with persistent headache had anxiety and 70.58% had depression. Of the patients with dyspnea, 76.47% had anxiety and 64.70% had depression.

### 3.2. Neuropsychological Findings

Neuropsychological characteristics are described in Table 2. The scores for each test are expressed as T score.

#### 3.2.1. Neuropsychological Results Depending on the Severity of the Disease

Table 3 shows the average performance in the different neuropsychological subtests depending on the severity of the disease. No significant differences were observed in the assessments at 6 months between groups considering the severity of the disease (ANOVA F between 0.14 and 3.11; *p* > 0.05).

#### 3.2.2. Neuropsychological Results Based on Cognitive Complaints

The sample was divided according to the subjects who presented with cognitive complaints (n = 38) and those who did not (n = 70). Table 4 shows the average neuropsychological performance in the different subtests depending on the SCC.

In the comparison at 6 months between both groups, significant differences were observed only in the TAVEC-B, with a better performance in the group with SCC (TAVEC-B: 6.19 vs. 5.11) (t: −3.36; *p* = 0.001) and higher rates of anxiety (HAD-A: 9.76 vs. 6.26) (t: −4.15; *p* = 0.001) and depression (HAD-D: 6.97 vs. 4.10) (t: −3.47; *p* = 0.001).

#### 3.2.3. Neuropsychological Results Based on Clinical Symptoms

##### Neuropsychological Results Depending on the Initial Symptom

It was studied whether there was a relationship between neurologic symptoms (headache, anosmia, and dysgeusia) at the beginning of the disease and a worse neuropsychological performance at 6 months (regardless of the persistence of the symptom at 6 months).

For patients with initial headache (n = 75) or anosmia (n = 54), no statistically significant differences were observed in the neuropsychological performance at 6 months (*p* > 0.05).

For patients with initial dysgeusia (N = 58), statistically significant differences were observed at 6 months, with a worse performance for patients who had presented this symptom in: TAVEC-1 with an average of 7.28 (SD: 1.58) vs. 8.00 (SD: 2.14) (ANOVA F: 2.56, *p* = 0.047); WMS-RI with a mean of 35.34 (SD: 5.49) vs. 37.49 (SD: 4.53) (ANOVA F: 4.42, *p* = 0.037); Letters and Numbers with an average of 9.26 (SD: 2.35) vs. 10.39 (SD: 2.18) (ANOVA F: 0.78, *p* = 0.012); and in BNT with an average of 50.36 (SD: 7.18) vs. 53.06 (SD: 4.98) (ANOVA F: 4.79, *p* = 0.024).

##### Neuropsychological Results Depending on the Persistent Symptom

For the most frequently reported persistent symptoms (dyspnea, headache, fatigue, anxiety, and depression), we studied whether there were significant differences in cognitive performance between patients.

Persistent dyspnea was associated with greater anxiety, with a mean of 10.35 (SD: 4.56) vs. 6.88 (SD: 4.23) (t: −3.00, *p* = 0.003), and greater depression, with an average of 7.65 (SD: 4.32) vs. 4.56 (SD: 4.12) (t: −2.81, *p* = 0.006).

Persistent headache was associated with greater anxiety, with a mean of 10.65 (SD: 3.63) vs. 6.82 points (SD: 4.34) (t: −3.40, *p* = 0.001), and depression, with an average of 8.06 (4.40) vs. 4.48 (SD: 4.04) (t:−3.30, *p* = 0.001).

Persistent fatigue was associated with a lower neuropsychological performance in tests of visual memory, attention, complex working memory, cognitive flexibility, processing speed, and language, and higher scores in anxiety and depression. The scores for the tests with statistically significant changes are represented in Table 5. No differences were observed in terms of ferritin values (t = 0.817, *p* = 0.41) or D-Dimer (t = 1.40, *p* = 0.16) at the onset of disease for patients with persistent fatigue.

Persistent anxiety was associated with a lower neuropsychological performance than individuals without this symptomatology in most neurocognitive areas evaluated, and persistent depression was associated with a lower neuropsychological performance on all administered subtests, except TAVEC-B and Reverse Digits (Table 5).

Patients with persistent fatigue and no cognitive complaints (N = 14) were compared with patients with persistent fatigue and cognitive complaints (N = 28). No significant differences were observed in neuropsychological performance, except in HAD-Anxiety: mean: 7.64 (SD: 3.77) vs. 10.68 (SD: 4.21) (t: −2.27, *p* = 0.028) and in HAD-Depression: mean: 5.00 (OF: 4.15) vs. 8.11 (4.62) (t: −2.12, *p* = 0.40), with the scores being higher for SCC patients.

We analyzed whether there were significant differences between males (N = 17) and females (N = 25) with persistent fatigue, between males (N = 22) with persistent anxiety and females (N = 27), and between males (N = 15) with persistent depression and females (N = 23). The results for the subtests with significant differences are shown in Table 6. Overall, women showed a lower neurocognitive performance in all neuropsychological subtests.

Table 7 shows the number of subjects and percentage for each neuropsychological subtest following the normal distribution based on SCC, persistent fatigue (symptom associated with cognitive alterations), and ICU admission (higher severity of illness).

## 4. Discussion

Our study was designed to characterize the extent of cognitive impairment and subjective cognitive complaints in patients 6 months after COVID-19 infection. The presence of persistent symptoms was frequent in our sample, consistent with the current literature, where a high percentage of individuals present symptoms that can last from weeks to months [16,17,18]. Neurological manifestations and brain fog have been reported in previous pandemics such as SARS and MERS-CoV [91]. The most frequently observed persistent symptom in our study was fatigue, followed by dyspnea and headache. These results are in line with those reported by various studies and meta-analyses carried out in the post-COVID-19 phase, where fatigue is the most frequently reported symptom [26,92,93,94,95,96,97] followed by dyspnea, headache, sleep disturbances, and neurocognitive and psychopathological alterations [19,20,21,30,31,32].

Cognitive complaints (35.2%), anxiety (45.37%), and depression (35.18%) were persistent symptoms 6 months after the infection. Neurocognitive and psychopathological alterations tend to persist for longer periods [22], with neurocognitive impairment and psychopathological symptoms, together with anxiety and depression [26], being the most prevalent after 6 and 12 months [23,33]. This leads to a hypothesis that the persistence of long-term symptoms of anxiety and depression could contribute to observed cognitive complaints.

In our study, females had higher rates of SCC [37], with persistent symptoms including anxiety and depression [98,99,100]. Although the risk factors for developing long COVID maybe diverse, other studies have already observed that the female sex could be a risk factor for developing post-COVID-19 fatigue [27,81] in both hospitalized [28] and non-hospitalized [29] women.

The above maybe due to a different inflammatory condition, associated with higher levels of IL-6 in women [101]. In our study, we did not find an association between ferritin and D-Dimer levels at the beginning of the disease with the development or maintenance of persistent fatigue.

### 4.1. Neuropsychological Outcomes

Neurocognitive impairment is one of the symptoms often referred to in post-COVID-19 syndrome. In our study, 35.2% of the evaluated subjects reported neuropsychological alterations 6 months after infection, as also reported by others [12].

#### 4.1.1. Illness Severity

We did not find an association between the severity of the disease and a worse neurocognitive performance. Other studies have also indicated no link between the disease’s severity and neuropsychological performance [10,37,38,39,40,41,42]. No neuropsychological differences were observed between the four groups considering severity, indicating that an association between disease severity and neurocognitive performance cannot be established. Others reported higher rates of neuropsychological alterations in COVID-19 participants [34,42,47]. However, patients with prior cognitive deficits were not excluded [39], and less specific batteries [35,36] and telephone/online assessments were often used [35,42] to report cognitive impairment. In our study, individuals were included regardless of their symptoms and disease severity, with an assessment battery designed specifically for this population. Examination was carried out face-to-face and discarded individuals with possible previous neurocognitive impairment, therefore avoiding confounding factors.

#### 4.1.2. Subjective Cognitive Complaints

As the severity of the disease was not related to neuropsychological performance, we checked if there were differences between those patients who reported cognitive complaints versus those who did not. There was no statistically significant difference between the performance of patients with and without cognitive complaints, except for anxiety and depression, being higher in the group with SCC. These results indicate that cognitive complaints are not associated with neurocognitive decline, but rather with the presence of anxiety–depressive symptoms. The later association has been described previously [5,10].

#### 4.1.3. Initial and Persistent Clinical Symptoms

Previous studies [5,10,38] have pointed out that neurological symptoms at the onset of the disease may be associated with a lower neurocognitive performance. We evaluated whether presenting with neurological symptoms in the acute phase of the disease had an impact on neuropsychology at 6 months. Our results indicated that the differences observed in the acute phase disappeared in the long term. This could be because, in the acute phase of the disease, patients are still in the recovery phase, and once the symptoms disappear, there is no impact on cognition. The only initial symptom associated with neurocognitive differences at 6 months was dysgeusia. This is expected, considering that the brain regions involved in the processing of taste (orbitofrontal cortex, cingulate gyrus, amygdala, hippocampus, and other areas of the limbic system) are the same as those affected in mild cognitive impairment, and neurodegenerative processes [94,95]. A recent systematic review and meta-analysis [96] concluded that taste abnormalities are common in various neurocognitive processes due to different etiologies. In addition, taste dysfunction was differentially associated with the severity of neurocognitive impairment, suggesting that taste dysfunction could represent a potential biomarker of neuropsychological impairment.

Considering the persistence of symptoms at 6 months, no differences were observed between the patients with headache or dyspnea in any neuropsychological test. Patients with these symptoms tend to obtain higher scores on anxiety and depression scales, observing again that patients with persistent symptoms present greater psychopathological symptomatology. On the contrary, a relationship was observed between fatigue, anxiety, and persistent depression with a lower neuropsychological performance.

Concern has been raised that SARS-CoV-2 has the potential to trigger a post-viral fatigue syndrome [24,25]. Our results point to this direction, given that persistent fatigue is associated with a poor neuropsychological performance in practically all cognitive areas: visual memory, attention, working memory, processing speed, cognitive flexibility, and language. Anxiety and depression were persistent symptoms associated with a worse neuropsychological performance on all administered subtests, and high rates were observed associated with fatigue [5,10].

When patients with persistent fatigue and cognitive complaints were compared to patients with persistent fatigue without cognitive complaints, no differences were found in neuropsychological performance, once again indicating there is no association between cognitive complaints and cognitive performance.

There was a higher prevalence of cognitive complaints in the female gender. We compared the neuropsychological performance between males and females with persistent fatigue, anxiety, and depression. Females obtained a lower overall performance on tests of visual memory, attention, working memory, processing speed, executive function, and language. Females had higher levels of anxiety and depression. Our study points to the existence of a strong relationship between psychopathological alterations, the persistence of symptoms, and cognitive complaints, and all these factors are closely associated with the female gender.

Considering that there were neuropsychological differences between patients with and without persistent fatigue, depression, and anxiety at 6 months, we hypothesize that, although the cognitive complaint does not translate into an established neurocognitive impairment, there could be a decline with respect to the previous level in individuals with these characteristics, mainly in tests of processing speed and executive function [48].

We aimed to determine if there was a greater number of subjects performing between one and two deviations below normal. We analyzed if there were differences between the subjects presenting with SCC, patients with an initial greater disease severity (ICU), and in patients with the persistent symptom mostly associated with a worse neurocognitive performance (fatigue).

As shown in Table 7, there were no individuals presenting cognitive impairment at two SD below normal, independent of the severity of the disease and cognitive complaints. We observed a higher percentage of subjects between one and two SD below in tests of processing speed. More specifically, in the Stroop test, a worse overall performance was obtained in all groups, while in the SDMT and TMT-A and B, the differences were seen between subjects without complaints (with expected percentages within normality) and the rest of the groups (with a greater percentage of individuals). We think that these tests could be the most sensitive in identifying cognitive changes in this population, since they are the ones that would allow us to differentiate between subjects without cognitive complaints and subjects with fatigue, ICU, or SCC. These results are in line with those of Ariza et al. [44], in which SDMT, MoCA, and phonetic fluency were observed to better discriminate patients with long COVID-19 from control subjects. Further, patients with SCC and fatigue obtained a lower performance on the test of attention/immediate memory (direct digits), as observed in patients with fibromyalgia, chronic fatigue, and cognitive complaints [102].

Our study was designed to evaluate the possible impact of COVID-19 infection on neurocognitive performance. Understanding the potential mechanisms of the pathogenesis of the various viral agents is essential for developing preventive methods and early treatments to prevent post-viral fatigue. The results of our study, without clear evidence of neurocognitive impairment, cast doubt on the impact of direct virus damage to the cortex and emphasize the need to understand the mechanisms by which fatigue persistence occur and the involvement of psychopathological symptoms such as anxiety and depression in the neurocognitive impact and quality of life of these individuals. Considering our results, clinicians should consider conducting an evaluation and treatment of psychoaffective symptomatology prior to cognitive evaluation. Neuropsychological cognitive stimulation should focus on daily life strategies in the young adult population, enhancing processing speed, planning capacity, and the use of external strategies.

## 5. Limitations

These results must be considered within the limitations of this study. Mainly, the lack of a previous neuropsychological assessment that could allow for observing minor differences in patients who did not present an established neurocognitive impairment but who presented a decline as compared to their baseline, as well as the lack of a control group that was age and gender matched. Including an ideal control group (without COVID-19 infection) was difficult given the state of the pandemic. We attempted to address this limitation through standardized and validated tests in our population and with strict inclusion and exclusion criteria. Serial assessments would provide valuable information on the evolution of neurocognitive deficits over time. We would welcome extending the study to a multicenter level, aiming to extrapolate the results to other populations and increase the external validity of the results. Another limitation is the small sample of the NH group, which could cause bias in the statistical analyses. However, it is important to emphasize that the largest sample size was found in the group with greater severity of the disease. Future studies should obtain larger samples of non-hospitalized patients for comparisons between groups. Both clinical symptoms at the beginning of the disease, in the longitudinal follow-up, and cognitive complaints were collected through open-ended questions with yes/no answers. It would be advisable for future studies to use standardized questionnaires to record the different symptomatology reported. Future studies should include more specific questionnaires to assess emotional functioning, neuropsychiatric symptoms, and PTSD, as well as sleep disturbances, which were related to a poor neurocognitive performance. Subsequently, also the presence of fatigue was assessed through an open question. Standardized measures, as well as a baseline and longitudinal timepoint assessments, would allow for a more comprehensive evaluation of fatigue severity and its impact on daily functioning and help to establish the specific impact of COVID-19 on fatigue. When correlating the data with biomarkers, we only used those that we were allowed to obtain as a routine. A limitation that follows considering the results of our study, in which fatigue is a frequent persistent symptom, is the lack of other specific inflammatory biomarkers such as IL-6, associated with systemic inflammation, which has been widely implicated in the presence of fatigue and persistent symptoms in these types of patients.

## 6. Conclusions

The findings in the current study allow us to characterize the cognitive profile of patients at 6 months after COVID-19 infection. Although neurocognitive impairment was not confirmed as a factor of the disease severity in the post-acute phase of the infection, the subjects tended to appear in a lower performance range, specifically on tests of processing speed, executive function, attention, and working memory. Persistent symptoms were common regardless of disease severity, mainly headache, dyspnea, and fatigue. Both symptoms and cognitive complaints were closely associated with higher rates of anxiety and depression and predominated in the female gender. SCC was not related with objective cognition, but with anxiety and depression symptoms. Individuals with persistent fatigue, anxiety, and depression had a worse neuropsychological performance at 6 months compared to individuals without these symptoms.

## Figures and Tables

**Table 1 brainsci-14-00604-t001:** Distribution of patients according to the severity of the disease and cognitive complaints.

SCC/Group Severity	NH	HOSPI	OXY	ICU
Without SCC (n = 70)	n = 7 (10%)	n = 16 (22.9%)	n = 33 (47.1%)	n = 14 (20%)
With SCC (n = 38)	n = 3 (7.9%)	n = 5 (13.2%)	n = 23 (60.5%)	n = 7 (18.4%)

NH: Not Hospitalized; HOSPI: Hospitalized, Not ICU, Not Oxygen; OXY: Hospitalized, Not ICU, Oxygen; ICU: ICU required Intensive Care Unit; SCC: Subjective Cognitive Complaints.

**Table 2 brainsci-14-00604-t002:** Neuropsychological performance at 6 months for all sample.

Neuropsychological Tests	All Sample Mean (SD)
TAVEC-1	51.49 (9.29)
TAVECTotal	54.35 (9.40)
TAVEC-B	45.92 (7.97)
TAVEC-IMR	54.25 (10.15)
TAVEC-IMRSC	55.27 (10.08)
TAVEC-DFR	53.92 (10.79)
TAVEC-DFRSC	54.67 (10.12)
TAVEC-REC	54.95 (6.66)
WMS-IMR	48.71 (7.12)
WMS-DFR	51.49 (6.72)
Digits Forward	47.93 (7.13)
Digits Backwards	49.05 (5.89)
Letter & Numbers	46.36 (6.33)
TMT-A	47.36 (8.68)
TMT-B	43.99 (8.43)
SDMT	44.23 (6.71)
Stroop Lecture	44.08 (7.63)
Stroop Color	43.75 (6.65)
Stroop Int.	43.89 (8.14)
Semantic Fluency	48.68 (8.24)
Phonemic Fluency	44.76 (6.89)
FCRO copy	52.63 (9.56)
BNT	48.34 (8.81)

TAVEC-1, Test de Aprendizaje Verbal España-Complutense learning 1; TavecTotal, Test de Aprendizaje Verbal España-Complutense sum of learning; TAVEC-B, Test de Aprendizaje Verbal España-Complutense learning B; TAVEC-IMR, Test de Aprendizaje Verbal España-Complutense Immediate Recall; TAVEC-IMRSC, Test de Aprendizaje Verbal España-Complutense Immediate Recall Semantic Clue; TAVEC-DFR, Test de Aprendizaje Verbal España-Complutense Deferred Free Recall; TAVEC-DFRSC, Test de Aprendizaje Verbal España-Complutense Deferred Free Recall Semantic Clue; TAVEC-REC, Test de Aprendizaje Verbal España-Complutense Recognition; WMS-IMR, Visual Reproduction of the Wechsler Memory Scale–IV Immediate Recall; WMS-DFR, Visual Reproduction of the Wechsler Memory Scale–IV Deferred Free Recall; TMT-A, Trail Making Test A; TMT-B, Trail Making Test B; SDMT, Symbol Digit Modalities Test; Stroop Int., Interference; FCRO, Complex Figure of Rey-Osterrieth; BNT, Boston Naming Test.

**Table 3 brainsci-14-00604-t003:** Neuropsychological performance at 6 months depending on the illness severity.

Neuropsychological Tests	NH (n = 10) Mean (SD)	HOSPI (n = 21) Mean (SD)	OXY (n = 56) Mean (SD)	ICU (n = 21) Mean (SD)
TAVEC-1	53.33 (8.66)	54.28 (9.25)	50.90 (9.67)	49.04 (8.30)
TAVECTotal	58.88 (10.54)	56.66 (9.12)	53.57 (9.42)	51.90 (8.72)
TAVEC-B	45.55 (8.81)	50.95 (7.68)	43.92 (6.51)	46.19 (9.73)
TAVEC-IMR	56.66 (7.07)	55.71 (10.28)	53.39 (10.66)	53.80 (10.23)
TAVEC-IMRSC	58.88 (6.00)	53.80 (8.04)	54.82 (11.75)	56.19 (8.64)
TAVEC-DFR	55.55 (5.27)	56.00 (7.53)	53.39 (12.68)	52.38 (9.95)
TAVEC-DFRSC	57.77 (4.40)	55.00 (6.88)	54.28 (11.88)	53.80 (9.73)
TAVEC-REC	55.55 (7.26)	51.50 (7.45)	55.63 (6.60)	56.00 (5.02)
WMS-IMR	49.72 (8.23)	48.92 (8.38)	48.77 (6.64)	48.21 (7.03)
WMS-DFR	51.11 (9.10)	53.21 (7.50)	51.44 (6.36)	50.00 (5.91)
Dígits Forward	50.00 (6.49)	48.21 (6.38)	47.48 (7.79)	47.73 (6.65)
Dígits Backward	51.11 (6.97)	49.28 (5.65)	49.06 (5.96)	47.97 (5.78)
Letter & Numbers	46.38 (3.56)	45.35 (5.93)	47.00 (7.22)	45.59 (5.29)
TMT-A	47.77 (5.06)	48.57 (7.76)	47.36 (9.64)	45.83 (8.52)
TMT-B	44.16 (4.67)	46.66 (8.19)	42.90 (9.49)	44.04 (7.00)
SDMT	44.44 (4.10)	44.52 (5.73)	44.41 (7.22)	43.69 (7.40)
Stroop Lecture	43.88 (6.13)	45.35 (7.83)	44.40 (8.03)	41.78 (6.98)
Stroop Color	42.22 (6.54)	44.88 (6.34)	43.87 (7.26)	42.87 (5.51)
Stroop Int.	42.77 (7.75)	46.42 (7.00)	44.27 (7.87)	40.62 (9.69)
Semantic Fluency	47.50 (5.15)	48.80 (6.73)	49.60 (8.97)	45.83 (7.83)
Fonetic Fluency	45.00 (3.95)	44.52 (6.45)	45.00 (7.16)	43.33 (6.48)
FCRO copy	57.44 (9.42)	55.00 (10.42)	51.65 (9.58)	50.95 (8.38)
BNT	50.77 (9.86)	47.59 (7.48)	48.48 (9.53)	47.38 (8.04)

TAVEC-1, Test de Aprendizaje Verbal España-Complutense learning 1; TavecTotal, Test de Aprendizaje Verbal España-Complutense sum of learning; TAVEC-B, Test de Aprendizaje Verbal España-Complutense learning B; TAVEC-IMR, Test de Aprendizaje Verbal España-Complutense Immediate Recall; TAVEC-IMRSC, Test de Aprendizaje Verbal España-Complutense Immediate Recall Semantic Clue; TAVEC-DFR, Test de Aprendizaje Verbal España-Complutense Deferred Free Recall; TAVEC-DFRSC, Test de Aprendizaje Verbal España-Complutense Deferred Free Recall Semantic Clue; TAVEC-REC, Test de Aprendizaje Verbal España-Complutense Recognition; WMS-IMR, Visual Reproduction of the Wechsler Memory Scale–IV Immediate Recall; WMS-DFR, Visual Reproduction of the Wechsler Memory Scale–IV Deferred Free Recall; TMT-A, Trail Making Test A; TMT-B, Trail Making Test B; SDMT, Symbol Digit Modalities Test; FCRO, Complex Figure of Rey-Osterrieth; BNT, Boston Naming Test.

**Table 4 brainsci-14-00604-t004:** Neuropsychological performance at 6 months based on subjective cognitive complaints.

Neuropsychological Tests	Without SCC (n = 70) Mean (SD)	With SCC (n = 38) Mean (SD)
TAVEC-1	52.17 (9.05)	46.57 (9.66)
TAVECTotal	58.97 (8.13)	49.21 (8.18)
TAVEC-B	45.71 (8.43)	46.57 (8.78)
TAVEC-IMR	56.57 (9.15)	47.63 (10.80)
TAVEC-IMRSC	56.85 (9.56)	48.42 (10.27)
TAVEC-DFR	56.14 (9.21)	49.21 (11.71)
TAVEC-DFRSC	56.14 (9.52)	47.10 (11.36)
TAVEC-REC	56.23 (5.45)	50.52 (9.57)
WMS-IMR	49.81 (6.72)	44.60 (6.24)
WMS-DFR	52.38 (6.71)	45.72 (7.25)
Dígits Forward	49.37 (6.76)	45.78 (6.31)
Dígits Backward	48.92 (5.78)	45.72 (7.18)
Letter & Numbers	46.89 (6.68)	43.94 (5.12)
TMT-A	48.46 (8.62)	45.46 (7.16)
TMT-B	44.73 (8.96)	42.63 (7.26)
SDMT	44.85 (7.12)	41.77 (5.38)
Stroop Lecture	46.01 (7.57)	42.30 (7.15)
Stroop Color	44.77 (6.74)	43.48 (4.66)
Stroop Int.	44.37 (8.88)	44.01 (5.11)
Semantic Fluency	49.97 (8.32)	46.64 (6.65)
Fonetic Fluency	45.17 (7.08)	42.82 (6.21)
FCRO copy	53.00 (9.39)	51.05 (9.59)
BNT	48.67 (9.19)	45.78 (7.33)

TAVEC-1, Test de Aprendizaje Verbal España-Complutense learning 1; TavecTotal, Test de Aprendizaje Verbal España-Complutense sum of learning; TAVEC-B, Test de Aprendizaje Verbal España-Complutense learning B; TAVEC-IMR, Test de Aprendizaje Verbal España-Complutense Immediate Recall; TAVEC-IMRSC, Test de Aprendizaje Verbal España-Complutense Immediate Recall Semantic Clue; TAVEC-DFR, Test de Aprendizaje Verbal España-Complutense Deferred Free Recall; TAVEC-DFRSC, Test de Aprendizaje Verbal España-Complutense Deferred Free Recall Semantic Clue; TAVEC-REC, Test de Aprendizaje Verbal España-Complutense Recognition; WMS-IMR, Visual Reproduction of the Wechsler Memory Scale–IV Immediate Recall; WMS-DFR, Visual Reproduction of the Wechsler Memory Scale–IV Deferred Free Recall; TMT-A, Trail Making Test A; TMT-B, Trail Making Test B; SDMT, Symbol Digit Modalities Test; BNT, FCRO, Complex Figure of Rey-Osterrieth; Boston Naming Test.

**Table 5 brainsci-14-00604-t005:** Differences in neuropsychological performance as a function of persistent symptoms.

NPS Tests—Symptoms	Fatigue No/Yes (n = 42)	Anxiety No/Yes (n = 49)	Depression No/Yes (n = 38)
TAVEC-1	-	-	7.91 (1.90)/7.08 (1–74) * d = 0.45
TAVECTotal	-	58.90 (8.47)/55.22 (10.43) * d = 0.38	59.36 (8.37)/53.32 (10.41) *** d = 0.63
TAVEC-IMR	-	-	12.79 (2.46)/11.29 (3.09) ** d = 0.53
TAVEC-IMRSC	-	-	13.80 (2.14)/12.37 (2.92) ** d = 0.55
TAVEC-DFR	-	-	13.11 (2.47)/11.55 (3.47) * d = 0.51
TAVEC-DFRSC	-	-	13.89 (1.94)/12.29 (3.12) ** d = 0.61
TAVEC-REC	-	15.51 (0.75)/14.90 (1.74) * d = 0.45	15.50 (0.77)/14.74 (1.89) * d = 0.52
WMS-IMR	37.20 (5.19)/34.95 (5.50) * d = 0.42	37.75 (4.26)/34.61 (6.13) ** d = 0.59	37.74 (4.49)/33.71 (6.00) *** d = 0.76
WMS-DRF	32.71 (7.60)/28.78 (8.58) * d = 0.48	33.22 (7.03)/28.82 (8.84) ** d = 0.55	33.29 (7.03)/27.42 (8.84) *** d = 0.73
Digits Forward	6.15 (1.15)/5.45 (1.21) ** d = 0.59	6.17 (1.06)/5.53 (1.30) ** d = 0.53	6.16 (1.13)/5.37 (1.21) *** d = 0.67
Letter and Number	10.17 (2.05)/9.19 (2.63) * d = 0.41	-	10.27 (2.13)/8.89 (2.44) ** d = 0.60
TMT-A	31.68 (12.99)/40.67 (21.15) ** d = 0.51	29.81 (10.76)/41.63 (20.88) *** d = 0.71	31.39 (14.57)/42.16 (19.36) ** d = 0.62
TMT-B	80.28 (38.94)/108.33 (71.02) * d = 0.48	79.80 (39.53)/105.28 (67.63) * d = 0.45	81.53 (45.41)/109.83 (66.93) * d = 0.49
SDMT	-	47.90 (10.10)/40.69 (13.62) ** d = 0.60	47.84 (9.75)/38.71 (14.33) *** d = 0.74
Stroop Lecture	103.26 (18.10)/89.02 (21.82) *** d = 0.71	104.58 (16.67)/89.23 (22.28) *** d = 0.78	104.53 (16.92)/84.70 (21.32) *** d = 1.03
Stroop Color	67.42 (11.87)/60.88 (12.70) ** d = 0.53	68.81 (11.08)/60.02 (12.67) *** d = 0.73	67.87 (10.76)/59.16 (13.81) *** d = 0.70
Stroop Int.	40.50 (10.77)/36.14 (10.72) * d = 0.40	41.50 (10.66)/35.48 (10.40) ** d = 0.57	40.83 (10.65)/34.95 (10.49) ** d = 0.55
Semantic Fluency	25.05 (6.74)/22.33 (5.78) * d = 0.43	25.63 (6.35)/22.02 (6.16) ** d = 0.57	25.47 (5.97)/21.26 (6.62) *** d = 0.66
Phonetic Fluency	16.00 (5.02)/13.81 (4.19) * d = 0.47	16.17 (4.73)/13.92 (4.67) ** d = 0.47	16.13 (4.53)/13.34 (4.86) ** d = 0.59
FCRO copy	-	-	33.81 (2.70)/31.34 (5.43) ** d = 0.57
BNT	-	52.78 (5.46)/50.29 (7.15) * d = 0.39	52.90 (5.23)/49.34 (7.63) * d = 0.54
HAD-Anxiety	6.00 (3.49)/9.67 (4.28) *** d = 0.93	4.14 (2.03)/11.39 (3.14) *** d = 2.39	5.26 (3.15)/11.42 (3.67) *** d = 1.80
HAD-Depression	3.76 (3.49)/7.07 (4.61) *** d = 0.80	2.25 (2.16)/8.41 (3.76) *** d = 2.00	2.33 (1.78)/10.05 (2.73) *** d = 3.34

TAVEC-1, Test de Aprendizaje Verbal España-Complutense learning 1; TavecTotal, Test de Aprendizaje Verbal España-Complutense sum of learning; TAVEC-IMR, Test de Aprendizaje Verbal España-Complutense Immediate Recall; TAVEC-IMRSC, Test de Aprendizaje Verbal España-Complutense Immediate Recall Semantic Clue; TAVEC-DFR, Test de Aprendizaje Verbal España-Complutense Deferred Free Recall; TAVEC-DFRSC, Test de Aprendizaje Verbal España-Complutense Deferred Free Recall Semantic Clue; TAVEC-REC, Test de Aprendizaje Verbal España-Complutense Recognition; WMS-IMR, Visual Reproduction of the Wechsler Memory Scale–IV Immediate Recall; WMS-DFR, Visual Reproduction of the Wechsler Memory Scale–IV Deferred Free Recall; TMT-A, Trail Making Test A; TMT-B, Trail Making Test B; SDMT, Symbol Digit Modalities Test; FCRO, Complex Figure of Rey-Osterrieth; BNT, Boston Naming Test; HAD, Hospital Anxiety and Depression scale; NPS, Neuropsychological; * *p* ≤ 0.05; ** *p* ≤ 0.01; *** *p* ≤ 0.001; d, Cohen’s effect.

**Table 6 brainsci-14-00604-t006:** Differences in neuropsychological performance as a function of persistent symptom and gender.

NPS Tests/Symptoms	Fatigue Male/Female	Anxiety Male/Female	Depression Male/Female
TAVEC-IMRSC	-	13.82 (2.40)/12.26 (2.85) * d = 0.59	-
WMS-IMR	37.94 (4.52)/32.95 (5.24) ** d = 1.01	37.09 (6.25)/32.59 (5.33) * d = 0.77	-
WMS-DFR	-	32.77 (8.51)/25.59 (7.86) ** d = 0.87	-
Digits Forward	6.12 (0.85)/5.00 (1.22) *** d = 1.06	6.32 (1.12)/4.89 (1.08) *** d = 1.29	6.00 (1.13)/4.96 (1.10) ** d = 0.93
Digits Backward	4.94 (1.14)/4.08 (1.03) * d = 0.79	-	-
Letter and Numbers	10.47 (2.12)/8.32 (2.62) ** d = 0.90	10.68 (2.19)/8.30 (2.35) *** d = 1.05	-
TMT-A	31.06 (9.18)/47.20 (24.46) * d = 0.87	33.36 (10.92)/48.37 (24.58) ** d = 0.78	34.80 (13.41)/46.96 (21.34) * d = 0.68
TMT-B	-	84.59 (38.66)/124.25 (82.46) * d = 0.61	-
Stroop Lecture	97.76 (18.50)/83.08 (22.23) * d = 0.71	98.68 (21.40)/81.23 (20.08) ** d = 0.84	-
Stroop Int.	40.76 (9.39)/33.00 (10.60) * d = 0.77	39.68 (10.46)/31.92 (9.10) ** d = 0.79	-
Semantic Fluency	-	24.41 (6.19)/20.07 (5.52) * d = 0.74	-
FCRO copy	34.14 (2.80)/31.34 (4.96) * d = 0.69	33.97 (3.11)/30.81 (5.20) * d = 0.73	-
BNT	52.76 (4.26)/48.24 (6.71) * d = 0.80	52.95 (5.58)/48.11 (7.63) * d = 0.72	-
HAD-Anxiety	-	10.05 (2.76)/12.48 (3.04) ** d = 0.83	9.47 (3.64)/12.70 (3.15) ** d = 0.94
HAD-Depression	5.00 (4.79)/8.84 (4.08) * d = 0.86	6.73 (3.66)/9.78 (3.30) ** d = 0.87	-

TAVEC-IMRSC, Test de Aprendizaje Verbal España-Complutense Immediate Recall Semantic Clue; WMS-IMR, Visual Reproduction of the Wechsler Memory Scale–IV Immediate Recall; WMS-DFR, Visual Reproduction of the Wechsler Memory Scale–IV Deferred Free Recall; TMT-A, Trail Making Test A; TMT-B, Trail Making Test B; Int., Interference; FCRO, Complex Figure of Rey-Osterrieth; BNT, Boston Naming Test; HAD, Hospital Anxiety and Depression scale; NPS, Neuropsychological; *, *p* ≤ 0.05; **, *p* ≤ 0.01; ***, *p* ≤ 0.001; d, Cohen’s effect.

**Table 7 brainsci-14-00604-t007:** Percentage and number of subjects following the normal distribution by subgroups.

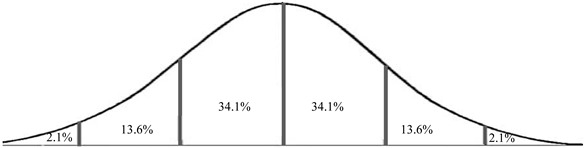						
NPS Tests	−2 (T < 30)	−1.5 (T = 30–39)	−1 (T = 40–49)	+1 (T = 50–59)	>1.5 (T = 60–69)	>+2 (T > 70)
**TAVEC-1**						
Without SCC		N = 1 (1.4%)	N = 13 (18.8%)	N = 31 (44.9%)	N = 18 (26.1%)	N = 6 (8.7%)
SCC		N = 1 (2.6%)	N = 10 (26.3%)	N = 18 (47.4%)	N = 5 (13.2%)	N = 4 (10.5%)
ICU		N = 1 (4.8%)	N = 4 (19%)	N = 13 (61.9%)	N = 2 (9.5%)	N = 1 (4.8%)
Fatigue			N = 9 (21.4%)	N = 22 (52.4%)	N = 7 (16.7%)	N = 4 (9.5%)
**TAVECTotal**						
Without SCC		N = 1 (1.4%)	N = 4 (5.7%)	N = 24 (15.7%)	N = 34 (48.6%)	N = 7 (10%)
SCC		N = 3 (7.9%)	N = 7 (18.4%)	N = 13 (34.2%)	N = 13 (34.2%)	N = 2 (5.3%)
ICU		N = 1 (4.8%)	N = 3 (14.3%)	N = 8 (38.2%)	N = 9 (42.9%)	
Fatigue		N = 3 (7.1%)	N = 4 (9.5%)	N = 15 (35.7%)	N = 18 (42.9%)	N = 2 (4.8%)
**TAVEC-B**						
Without SCC		N = 7 (10%)	N = 24 (34.3%)	N = 32 (45.7%)	N = 6 (8.7%)	N = 1 (1.4%)
SCC		N = 1 (2.6%)	N = 16 (42.1%)	N = 17 (44.7%)	N = 4 (10.5%)	
ICU		N = 3 (14.3%)	N = 6 (28.6%)	N = 8 (38.2%)	N = 4 (19%)	
Fatigue		N = 2 (4.8%)	N = 14 (33.3%)	N = 21 (50%)	N = 5 (11.9%)	
**TAVEC-IMR**						
Without SCC			N = 9 (12.9%)	N = 18 (25.7%)	N = 31 (20.3%)	N = 12 (17.1%)
SCC		N = 3 (7.9%)	N = 10 (26.3%)	N = 11 (28.9%)	N = 12 (31.6%)	N = 2 (5.3%)
ICU			N = 5 (23.8%)	N = 6 (28.6%)	N = 7 (33.3%)	N = 3 (14.3%)
Fatigue		N = 3 (7.1%)	N = 8 (19%)	N = 13 (31%)	N = 15 (35.7%)	N = 3 (7.1%)
**TAVEC-IMRSC**						
Without SCC		N = 1 (1.4%)	N = 6 (8.6%)	N = 22 (31.4%)	N = 26 (37.1%)	N = 15 (21.4%)
SCC		N = 3 (7.9%)	N = 5 (13.2%)	N = 13 (34.2%)	N = 14 (36.8%)	N = 3 (7.9%)
ICU			N = 1 (4.8%)	N = 10 (47.6%)	N = 6 (28.6%)	N = 4 (12.9%)
Fatigue		N = 1 (2.4%)	N = 5 (11.9%)	N = 14 (33.3%)	N = 16 (38.1%)	N = 6 (14.3%)
**TAVEC-DFR**						
Without SCC			N = 11 (15.7%)	N = 15 (21.4%)	N = 34 (48.6%)	N = 10 (14.3%)
SCC		N = 6 (15.8%)	N = 7 (18.4%)	N = 9 (23.7%)	N = 12 (31.6%)	N = 3 (7.9%)
ICU			N = 6 (28.6%)	N = 6 (28.6%)	N = 7 (33.3%)	N = 2 (9.5%)
Fatigue		N = 4 (9.5%)	N = 7 (16.7%)	N = 11 (26.2%)	N = 15 (35.7%)	N = 5 (11.9%)
**TAVEC-DFRSC**						
Without SCC		N = 3 (4.3%)	N = 4 (5.7%)	N = 20 (28.6%)	N = 33 (47.1%)	N = 10 (14.3%)
SCC		N = 3 (7.9%)	N = 6 (15.8%)	N = 12 (31.6%)	N = 13 (34.2%)	
ICU		N = 1 (4.8%)	N = 2 (9.5%)	N = 8 (38.2%)	N = 8 (38.2%)	N = 2 (9.5%)
Fatigue		N = 3 (7.1%)	N = 5 (11.9%)	N = 11 (26.2%)	N = 18 (42.9%)	N = 5 (11.9%)
**TAVEC-REC**						
Without SCC			N = 2 (2.9%)	N = 22 (31.9%)	N = 45 (65.2%)	
SCC		N = 2 (5.3%)	N = 2 (5.3%)	N = 17 (44.7%)	N = 15 (39.5%)	
ICU			N = 8 (25.8%)	N = 12 (38.7%)		
Fatigue			N = 2 (4.8%)	N = 20 (47.6%)	N = 20 (47.6%)	
**WMS-IMR**						
Without SCC		N = 3 (4.3%)	N = 28 (40.5%)	N = 35 (52.1%)	N = 2 (2.9%)	
SCC		N = 6 (15.8%)	N = 16 (42.1%)	N = 16 (42.1%)		
ICU		N = 1 (4.8%)	N = 10 (47.6%)	N = 10 (47.6%)		
Fatigue		N = 4 (9.5%)	N = 20 (47.6%)	N = 18 (42.9%)		
**WMS-DFR**						
Without SCC		N = 2 (2.9%)	N = 21 (30.3%)	N = 36 (52.1%)	N = 10 (14.3%)	
SCC			N = 17 (44.7%)	N = 16 (42.1%)	N = 5 (13.2%)	
ICU		N = 1 (4.8%)	N = 6 (28.6%)	N = 14 (66.6%)		
Fatigue		N = 1 (2.4%)	N = 18 (42.9%)	N = 20 (47.6%)	N = 3 (7.1%)	
**Digits Forward**						
Without SCC		N = 4 (5.7%)	N = 29 (41.5%)	N = 32 (45.7%)		
SCC		N = 8 (21.2%)	N = 19 (50%)	N = 11 (28.9%)		
ICU		N = 2 (9.5%)	N = 11 (52.4%)	N = 8 (38.2%)		
Fatigue		N = 7 (16.7%)	N = 21 (50%)	N = 14 (33.3%)		
**Digits Backward**						
Without SCC		N = 2 (2.9%)	N = 31 (44.3%)	N = 34 (48.7%)	N = 3 (4.3%)	
SCC			N = 19 (50%)	N = 17 (44.7%)	N = 2 (5.3%)	
ICU			N = 11 (52.4%)	N = 10 (47.6%)		
Fatigue		N = 1 (2.4%)	N = 22 (52.4%)	N = 17 (40.4%)	N = 2 (4.8%)	
**Letter and Numbers**						
Without SCC		N = 7 (10%)	N = 36 (52.1%)	N = 23 (32.9%)	N = 4 (5.7%)	
SCC		N = 4 (10.5%)	N = 26 (68.3%)	N = 8 (21.2%)		
ICU		N = 2 (9.5%)	N = 12 (57.1%)	N = 7 (33.3%)		
Fatigue		N = 6 (14.3%)	N = 26 (61.9%)	N = 10 (23.8%)		
**TMT-A**						
Without SCC		N = 7 (10%)	N = 32 (45.7%)	N = 20 (28.6%)	N = 11 (15.8%)	
SCC		N = 8 (21.2%)	N = 18 (47.4%)	N = 10 (26.3%)	N = 2 (5.3%)	
ICU		N = 5 (23.8%)	N = 9 (42.8%)	N = 5 (23.8%)	N = 2 (9.5%)	
Fatigue		N = 7 (16.7%)	N = 23 (54.7%)	N = 10 (23.8%)	N = 2 (4.8%)	
**TMT-B**						
Without SCC	N = 1 (2.4%)	N = 10 (14.3%)	N = 36 (52.1%)	N = 18 (26%)	N = 3 (4.3%)	
SCC		N = 12 (31.6%)	N = 18 (47.4%)	N = 5 (13.2%)	N = 1 (2.6%)	
ICU		N = 6 (28.5%)	N = 9 (42.8%)	N = 6 (28.6%)		
Fatigue	N = 1 (2.4%)	N = 12 (30%)	N = 21 (50%)	N = 6 (14.3%)	N = 1 (2.4%)	
**SMDT**						
Without SCC		N = 11 (15.7%)	N = 38 (54.2%)	N = 19 (27.3%)	N = 2 (5.3%)	
SCC		N = 10 (26.3%)	N = 22 (57.9%)	N = 5 (13.2%)	N = 1 (2.6%)	
ICU		N = 5 (23.8%)	N = 12 (57.1%)	N = 3 (14.4%)	N = 1 (4.8%)	
Fatigue		N = 8 (19%)	N = 30 (71.5%)	N = 3 (7.1%)	N = 1 (2.4%)	
**Stroop Lecture**						
Without SCC		N = 13 (18.8%)	N = 30 (43.4%)	N = 24 (24.7%)	N = 2 (5.3%)	
SCC		N = 17 (44.7%)	N = 17 (44.7%)	N = 3 (7.9%)	N = 1 (2.6%)	
ICU		N = 6 (28.5%)	N = 13 (61.9%)	N = 1 (4.8%)	N = 1 (4.8%)	
Fatigue		N = 16 (38.1%)	N = 22 (52.4%)	N = 3 (7.1%)	N = 1 (2.4%)	
**Stroop Color**						
Without SCC		N = 14 (20.6%)	N = 38 (55.9%)	N = 14 (20.6%)	N = 2 (5.3%)	
SCC		N = 11 (28.9%)	N = 23 (60.6%)	N = 4 (10.5%)		
ICU		N = 2 (9.5%)	N = 15 (75%)	N = 3 (15%)		
Fatigue		N = 11 (26.2%)	N = 29 (69%)	N = 2 (4.8%)		
**Stroop Int.**						
Without SCC		N = 18 (26%)	N = 29 (42.7%)	N = 18 (26%)	N = 3 (4.3%)	
SCC		N = 12 (31.4%)	N = 18 (47.4%)	N = 8 (21.2%)		
ICU		N = 6 (28.5%)	N = 13 (61.9%)	N = 1 (4.8%)		
Fatigue		N = 14 (33.3%)	N = 14 (33.3%)	N = 8 (19%)	N = 2 (4.8%)	
**Semantic Fluency**						
Without SCC		N = 8 (11.5%)	N = 20 (28.6%)	N = 34 (48.7%)	N = 7 (10%)	N = 1 (2.4%)
SCC		N = 7 (18.4%)	N = 19 (50%)	N = 10 (26.3%)	N = 2 (5.3%)	
ICU		N = 4 (19.1%)	N = 9 (42.9%)	N = 7 (33.3%)	N = 1 (4.8%)	
Fatigue		N = 6 (14.3%)	N = 20 (47.6%)	N = 14 (33.3%)	N = 2 (4.8%)	
**Phonetic Fluency**						
Sense queixes	N = 1 (2.4%)	N = 13 (18.8%)	N = 39 (55.8%)	N = 13 (18.8%)	N = 4 (5.7%)	
Queixes cognitives		N = 5 (23.7%)	N = 19 (50%)	N = 10 (26.3%)		
UCI		N = 5 (23.8%)	N = 12 (57.1%)	N = 4 (19.1%)		
Fatiga		N = 10 (23.8%)	N = 24 (57.2%)	N = 8 (19%)		
**FCRO copy**						
Without SCC		N = 2 (5.3%)	N = 27 (38.5%)	N = 19 (27.3%)	N = 22 (31.9%)	
SCC		N = 3 (7.9%)	N = 14 (36.9%)	N = 11 (28.9%)	N = 10 (26.3%)	
ICU		N = 1 (4.8%)	N = 9 (42.9%)	N = 7 (33.3%)	N = 4 (19.1%)	
Fatigue		N = 3 (7.1%)	N = 19 (45.3%)	N = 9 (21.4%)	N = 11 (26.2%)	
**BNT**						
Without SCC	N = 1 (2.4%)	N = 10 (14.3%)	N = 22 (31.9%)	N = 30 (43.4%)	N = 7 (10%)	
SCC		N = 3 (7.9%)	N = 19 (50%)	N = 12 (31.6%)	N = 4 (10.5%)	
ICU		N = 3 (9.6%)	N = 9 (42.9%)	N = 7 (33.3%)	N = 2 (9.5%)	
Fatigue		N = 6 (14.3%)	N = 22 (52.4%)	N = 12 (28.5%)	N = 2 (4.8%)	

TAVEC-1, Test de Aprendizaje Verbal España-Complutense learning 1; TavecTotal, Test de Aprendizaje Verbal España-Complutense sum of learning; TAVEC-B, Test de Aprendizaje Verbal España-Complutense learning B; TAVEC-IMR, Test de Aprendizaje Verbal España-Complutense Immediate Recall; TAVEC-IMRSC, Test de Aprendizaje Verbal España-Complutense Immediate Recall Semantic Clue; TAVEC-DFR, Test de Aprendizaje Verbal España-Complutense Deferred Free Recall; TAVEC-DFRSC, Test de Aprendizaje Verbal España-Complutense Deferred Free Recall Semantic Clue; TAVEC-REC, Test de Aprendizaje Verbal España-Complutense Recognition; WMS-IMR, Visual Reproduction of the Wechsler Memory Scale–IV Immediate Recall; WMS-DFR, Visual Reproduction of the Wechsler Memory Scale–IV Deferred Free Recall; TMT-A, Trail Making Test A; TMT-B, Trail Making Test B; SDMT, Symbol Digit Modalities Test; FCRO, Complex Figure of Rey-Osterrieth; BNT, Boston Naming Test; NPS, Neuropsychological.

## Data Availability

The data presented in this study are available on request from the corresponding author due to ethical reasons.

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
