# Peer review of "Neurocognitive and Neuropsychiatric Sequelae in Long COVID-19 Infection"

_brainsci, 2024, doi:10.3390/brainsci14060604_

Round 1

Reviewer 1 Report

Comments and Suggestions for Authors

  •  
  • Methods:

This section would benefit from a more comprehensive description of the materials used, including detailed information about the assessment items, example questions, and the scoring methods for each questionnaire.

Results:

To enhance the clarity and conciseness of this section, it is recommended to report effect sizes, such as Cohen's d. Additionally, only significant findings should be detailed in the text to avoid excessive verbosity. The inclusion of figures would also be advantageous.

Discussion:

A dedicated paragraph discussing the theoretical and practical implications of the study findings should be included to enrich the discussion section.

Author Response

Dear Reviewer, thank you very much for taking the time to review this manuscript. Thank you for your suggestions to improve our article, we appreciate the effort. Please find the detailed responses below and the corresponding revisions/corrections highlighted/in track changes in the re-submitted files.

Please find attach the document with the answers point by point.

Reviewer 2 Report

Comments and Suggestions for Authors

Dear Authors,

the Manuscript “Long Neurocognitive and Neuropsychiatric Sequelae in post-2 COVID-19 infection” is interesting, nevertheless major and minor revisions are necessary.

Please, delete all the data (SD, p values, Chi coefficient, etc) from the Abstract section.

Authors indicate different n values for NH, HOSP, OXY and ICU. In particular, NH n=10 may cause a bias in statistical analyses.

The main limitation of the study is the low number of Non Hospitalized patients enrolled. Please discuss it.

Title: Authors indicate in the title “in post-2 COVID-19 infection”, it would be appropriate to specify Long-COVID-19 as in the Abstract.

Line 39: Olfactory and gustatory deficits, like phantosmia, parosmia, phantogeusia, and parageusia, were recently studied by Ercoli et al., 2021 in Long COVID-19 patients. Please, include it.

All the manuscript: please unify the following terms: Chi2, Chi2, Chi square.

All the manuscript:  Please correct Long-COVID-19 instead of long-COVID.

Line 53: correct COVID-19.

Line 139: Authors need to indicate the software used for statistical analysis.

Line 149-153: indicate if these results were obtained regarding all the participants, or SCC, NH, etc..

Tables 5 and 6: Revise the tables and delete t coefficients. Moreover, indicate p values as symbol (*, **, ***) and write them in the legend, in order to obtain a better comprehension of the tables.

Line 273: insert comma as (N=27),

Table 7: Avoid colours in the table and in the data or explain them in legend.

Line 313: please insert % and comma after 45.37.

Line 441: Authors need to better explain the relationship between inflammatory biomarkers and patients with Long Covid.

Best regards.

Comments on the Quality of English Language

Minor editing of English language required

Author Response

Dear Reviewer, thank you very much for taking the time to review this manuscript. Thank you for your suggestions, we appreciate the effort. Please find the detailed responses below and the corresponding revisions/corrections highlighted/in track changes in the re-submitted files.

Reviewer 3 Report

Comments and Suggestions for Authors

Comments and suggestions:

  • The cross-sectional design at a single timepoint (6 months post-infection) does not allow evaluation of the longitudinal trajectory of symptoms. Serial assessments would provide valuable information on the evolution of neurocognitive deficits over time. Please comment.
  • The lack of a control group without COVID-19 makes it difficult to determine if the reported symptoms are higher than the background prevalence in the general population. An age and gender-matched control group would strengthen the findings.
  • The sample is from a single center in Spain, so the generalizability to other populations is uncertain. A multi-center study would increase the external validity.
  • More details could be provided on the neuropsychological tests administered and the specific scores used in the analyses (e.g. raw scores vs scaled scores, total scores vs subscores).
  • The discussion can further expand on the potential mechanisms underlying the "brain fog" and other neurocognitive symptoms in long COVID, and compare/contrast the findings with other post-viral syndromes.
  • Proofreading is needed for minor grammatical and stylistic issues.

Regarding the assessment of fatigue in this study, there are a few limitations that warrant discussion:

  1. Lack of a standardized fatigue questionnaire: The authors mention that the presence of fatigue was assessed through an open question to participants asking if they had noticed fatigue after the COVID-19 infection. While this provides some useful information, the use of a validated fatigue scale or questionnaire would have strengthened the assessment. Standardized measures such as the Fatigue Severity Scale, Chalder Fatigue Scale, or the Modified Fatigue Impact Scale are commonly used in research on post-viral fatigue and would allow for a more comprehensive evaluation of fatigue severity and its impact on daily functioning.
  2. Absence of baseline fatigue measurement: The study did not assess fatigue levels prior to the COVID-19 infection, making it difficult to determine if the reported fatigue is truly a new symptom or an exacerbation of pre-existing fatigue. Collecting information on baseline fatigue would help to establish the specific impact of COVID-19 on fatigue.
  3. Limited information on the characteristics of fatigue: The binary yes/no assessment of fatigue provides limited insight into the nature, duration, and variability of the fatigue symptoms. Collecting more detailed information, such as the timing of fatigue onset, daily fluctuations, and exacerbating/relieving factors, would paint a more comprehensive picture of the fatigue experience in these patients.
  4. No objective measures of fatigue: The study relied solely on self-reported fatigue, which can be influenced by various factors such as mood, sleep, and stress. Incorporating objective measures, such as actigraphy to assess physical activity levels or attention/vigilance tests to evaluate mental fatigue, would provide complementary data to the subjective reports.
  5. Lack of longitudinal assessment: Fatigue was only assessed at a single timepoint of 6 months post-infection. Monitoring fatigue levels over time, such as at 3, 6, and 12 months, would provide valuable information on the trajectory and persistence of fatigue symptoms.

Author Response

Dear Reviewer, thank you very much for taking the time to review this manuscript. Thank you for your suggestions to improve our article, we appreciate the effort. Please find the detailed responses below and the corresponding revisions/corrections highlighted/in track changes in the re-submitted files.

Round 2

Reviewer 2 Report

Comments and Suggestions for Authors

Dear Authors,

the Manuscript “Neurocognitive and Neuropsychiatric Sequelae in Long
-COVID-19 infection” has a good level of novelty.
Authors responded to the Report1 comments and implemented the manuscript. 

Best regards

Reviewer 3 Report

Comments and Suggestions for Authors

The authors responded to my comments very well. Thank you.